# Open Pyeloplasty in Infants under 1 Year—Proven or Meaningless?

**DOI:** 10.3390/children10020257

**Published:** 2023-01-31

**Authors:** Franziska Vauth, Pirmin Zöhrer, Florian Girtner, Wolfgang H. Rösch, Aybike Hofmann

**Affiliations:** Department of Pediatric Urology, Clinic St. Hedwig, University Medical Center Regensburg, 93049 Regensburg, Germany

**Keywords:** ureteropelvic junction obstruction, abdominal surgery, small infants, long-term outcome, parental satisfaction

## Abstract

The use of minimally invasive surgery (MIS) to treat ureteropelvic junction obstruction (UPJO) in children has significantly increased. Nowadays, open pyeloplasty (OP) seems to lose importance. The aim of this study is to evaluate the safety and efficacy of OP in infants < 1 year. Medical records of patients < 1 year with UPJO who had undergone retroperitoneal OP (January 2008–February 2022) at our institution were retrospectively analyzed. Included patients were operated on according to a modified Anderson–Hynes technique. All clinical, operative, and postoperative (1 month–5 years’ follow-up) data were collected. Additionally, a nonvalidated questionnaire was sent to the patients/parents. A total of 162 infants (124 boys) met the inclusion criteria. The median age at surgery was 3 months (range: 0–11 months). The median operation time was 106 min (range: 63–198 min). None of the patients had severe surgical complication (Clavien–Dindo > 3). The nonvalidated questionnaire showed a high impact of quality of life. Follow-up was in median 30.5 months (0–162 months). OP is still a reliable procedure with good long-term results especially in infants < 1 year of age, which can be performed in a variety of centers.

## 1. Introduction

The widespread availability and use of prenatal ultrasound of the urinary tract has caused an increase in the incidence of hydronephrosis in newborns [1,2]. Ureteropelvic junction obstruction (UPJO) is the most common cause of prenatal hydronephrosis [2,3]. Postnatally untreated, it can cause a severe and permanent loss of renal function. Despite a still ongoing discussion about the correct follow-up and timing of surgery [4,5,6], there are a considerable number of infants under 1 year who need surgical treatment.

Open pyeloplasty has long been the gold standard for the operative management of UPJO in children since it was first described 1949 by Anderson and Hynes [7,8]. With the further development of laparoscopic procedures and lately robot-assisted procedures in pediatric pyeloplasty, they have gained popularity. While there are significant data comparing the objective aspects of open pyeloplasty and robot-assisted laparoscopic pyeloplasty (RALP) in infants and adolescents [9,10,11,12], there is a shortage of data for very young infants and also data for assessing patient (and parents) satisfaction [13,14].

The arguments in favor of laparoscopic procedures or RALP are less scaring, lower pain scales and shorter inpatient stay [10,11]. Nevertheless, its acceptance in pediatrics was negatively affected by longer operating times, smaller working space, and limited fine surgical instruments [15,16]. At the same time, Sorensen et al. showed a short learning curve for RALP. After 15 to 20 robotic cases, overall operative times for RALP cases were consistently within 1 SD of the average open pyeloplasty time with no significant difference in overall operative time [17].

Although this minimally invasive technique affords significant advantages to the patient over the open approach, it is performed by only few surgeons, limiting its use to high-volume centers [3].

In this study, we aim to evaluate whether open pyeloplasty is still a safe and reliable procedure with good cosmetic results especially in infants under 1 year of age.

## 2. Materials and Methods

We identified 368 children who had undergone open pyeloplasty between January 2008 and February 2022 at our department (full member of ERN eUROGEN). A total of 185 children (50.3%) were under 1 year of age at the time of surgery. Patients with other cause of hydronephrosis than ureteropelvic junction obstruction were excluded (e.g., secondary stenosis due to primary megaureter), or patients who underwent other surgery in the same term (e.g., pyeloplasty and circumcision). We also excluded patients with skin incision other than described below. A total of 162 children met the inclusion parameters, 124 boys (76.7%) and 38 girls (23.3%).

The medical records were retrospectively reviewed in terms of demographic data, operation time, level of experience of the surgeon, length of hospital stay, duration of stent placement, febrile urinary tract infection during ureteral stenting, ultrasound findings (grade of hydronephrosis according to a consensus group pediatric nephrology working society [18], anteroposterior pelvic diameter (APD), parenchyma thickness) before and after surgery in a standardized follow-up, separate renal function (SRF) in MAG3 as well as analgesia during hospital stay, intraoperative and postoperative complications according to Clavien–Dindo (CD), and inpatient readmission within 30 days. We also examined the need of a redo pyeloplasty.

Complications were regarded as any deviation from the expected postoperative course according to the five-grade Clavien–Dindo classification [19].

In the second part of this study, a nonvalidated self-designed questionnaire (in German language) was sent in September 2022 to all included patients by mail to evaluate the postoperative course from the parents’ point of view, as well as the satisfaction in the long-term course. Additionally, they were asked to take a photo of the scar alongside a metric ruler to obtain scar length values. The parents were asked to return the completed questionnaire and the photo of the scar either by mail (prepaid envelope enclosed) or by e-mail within 4 weeks (Appendix A).

In the third part, these photos were independently categorized by 2 experienced surgeons using a modified Vancouver Scar Scale [20]. The scale was modified for usage on photographs and linear scars. Instead of measuring the height, it was described as flat, slightly raised, fairly raised, bulging, and sunken. Pliability was left out as it is not representable on photographs. The higher the scoring, the worse the cosmesis, whereby the maximum score of 9 reflects the worst imaginable scar. The lowest scores (0–2) reflect the best imaginable scar or almost normal skin (Appendix A).

### 2.1. Surgical Management

#### 2.1.1. Preoperative Management

Newborns with hydronephrosis, prenatally or postnatally diagnosed, underwent further examination with repeated ultrasound and MAG3 scintigraphy. If an obstructive drainage was detected, pyeloplasty was recommended according to the EAU guidelines [21].

#### 2.1.2. Surgical Technique

All procedures were performed with a retroperitoneal approach and a transverse abdominal alternating incision.

After insertion of a transurethral bladder catheter and supine positioning of the patient, a transverse incision is made over the course of the rectus abdominis muscle on the affected side (Figure 1a). The anterior muscle fascia is opened longitudinally from the lateral side, and the muscle is retracted without disconnecting it. Then the posterior muscle fascia is opened, the retroperitoneal space is widened, and the peritoneum is bluntly displaced medially. Gerota’s fascia is incised and once the ureteropelvic junction has been exposed (Figure 1b) fine stay sutures are placed in the anterior portion of the upper ureter and the anterior portion of the renal pelvis, and both structures are gently mobilized. V-like transection of the renal pelvis leaves a caudal flap. The ureter is then spatulated far beyond the obstruction to provide an open tube for triangulated anastomoses. The suture line is made with a combination of interrupted sutures at the most dependent portion of the ureter and caudal flap of the renal pelvis, and an uninterrupted suture on either side of the anastomoses, forming a wide funnel-shaped connection. Before the anastomoses are completed, a ureteral JJ stent (usually 4.8 Charr.) is inserted, and correct positioning is checked by reflux of urine from the bladder. Once the anastomoses has been completed, a Mini-Vac^®^ drain is placed near the area of the anastomoses, and the wound is closed in layers using absorbable suture (Figure 1c).

#### 2.1.3. Postoperative Management

As for short-term postoperative management, the drainage was removed after 2–3 days, depending on the output. The catheter stayed for 5 days to ensure a zero-pressure urine drainage. Generally, patients were dismissed 5 days after surgery.

All children received low-dose antibiotic prophylaxis (trimethoprim 2 mg/kg body weight per day) during stent placement. Ureteral stent removal was scheduled 6 weeks postoperatively, and patients needed a short anesthesia for cystoscopic removal of the stent.

The long-term postoperative standardized follow-up is as follows: ultrasound before surgery, 2 days after surgery during hospital stay with indwelling stent, 4–6 weeks after surgery in the context of stent removal, first control 3- to 6-months postoperative, and then after 1 year. After that, once a year for the next 5 years. All ultrasound examinations were performed by a pediatric urologist at our clinic. With stable conditions (complete reduction of dilatation or stable residual dilatation), ultrasound and clinical examination are performed every 2–3 years until the age of 18 years.

The success of surgery was defined as a remission or at least a significant reduction of the dilatation (decrease in APD and pelvis and caliceal tension decrease).

MAG3 scintigraphy was carried out in case of persistent high-grade dilatation or increase in dilatation during ultrasound at the earliest period of 6 months after surgery. There was no routine MAG3 scintigraphy conducted postoperatively.

In case of febrile urinary tract infection, criteria of EAU guidelines according to urine sampling and urine culture were applied [21].

### 2.2. Statistical Analysis

Continuous variables are shown as median (range: minimum–maximum), median (interquartile range), or mean (±standard deviation) as appropriate. Categorical variables are expressed as counts and percentages. Comparisons of non-normally distributed continuous variables between the groups were performed with the Wilcoxon signed rank test, and comparisons of categorical variables between groups with Fisher’s Exact x^2^ test. All analyses were conducted using SPSS^®^, version 29.0 (IBM Corp., Armonk, NY, USA).

### 2.3. Ethics Statement

Approval from the institutional ethics committee (no. 22-3003-101) and informed consent from all prospectively included participants were obtained.

## 3. Results

At our department, in 184 children under 1 year of age, open pyeloplasty was conducted between January 2008 and February 2022. A total of 22 patients had to be excluded because of a different surgical incision (6 patients), repyeloplasty (1 patient), and secondary ureteric pelvic junction obstruction (15 patients). Consequently, the records of 162 patients were reviewed for this study. In 13 (8%) patients, only surgical data could be obtained due to a home-based follow-up.

Open pyeloplasty was in median performed at an age of 3 months (0–11 months). The majority of the patients (77.9%) were under the age of 6 months. All patients had a weight ≤ 10 kg. Detailed patient characteristics are shown in Table 1.

All included patients were conducted with the same surgical technique. The median operative time was 106 min. Surgery was performed by multiple surgeons, and nearly half (49.4%) of the cases were carried out by fellows with the assistance of one experienced surgeon. Astonishingly, there was no significant difference regarding the median operative time between fellows (106.5 min (IQR 91–129 min)) and experienced surgeons (106.0 min (IQR 89.75–122.5 min)).

A JJ stent was placed in all patients for urinary drainage. In 6 (3.7%) patients, a ureteral stent could not be antegradely inserted. In those cases, a nephrostomy tube drainage was utilized. The duration of the urinary drainage was in median 42 days (IQR 40–46 days). (Table 2) A further intraoperative complication was respiratory difficulties in a former preterm infant, without requiring any further postoperative therapy. None of the patients required monitoring in an intensive care unit.

Postoperative complications according to the Clavien–Dindo (CD) classification were observed in 34 (21%) patients. A total of 2 patients had more than one complication. Febrile urinary tract infection (CD 2) was the most frequent minor complication (*n* = 16 (9.9%)). There were no CD 3a complications. A total of 17 (10.5%) patients required invasive treatment (CD 3b), predominantly related to stent displacement and consequent stent replacement under anesthesia. Additionally, 1 patient suffered urinoma with stent replacement. One redo pyeloplasty had to be performed 4 months after initial surgery due to a radiographically proven persistent pyeloureteral obstruction. None of the patients had relevant bleeding. Detailed data are shown in Table 3.

Postoperative pain management data were available for 161 (99.4%) patients, of whom 158 (97.5%) required postoperative pain medication. The median analgesic requirement was 42 h (IQR 24–60 h), with a median amount of 300 mg (IQR 80–530 mg) metamizol and 125 mg (IQR 75–240 mg) paracetamol. The median length of hospital stay was 5 days (range: 3–12 days).

A significant improvement of the median renal pelvic diameter from 22.0 mm before operation to 8.0 mm 1 year after surgery (*p* < 0.001), a renal calyx from 13.0 to 4 mm (*p* < 0.001), and parenchymal thickness from 4.9 to 9.0 mm (*p* < 0.001) could be shown (Table 4).

Despite the significant improvement in median, in 10 patients, MAG3 had to be initiated due to a persistence of grade 3–4 hydronephrosis (9 patients) and grade 2 hydronephrosis after postoperative initial complete remission (1 patient). None of these patients required redo pyeloplasty until now.

A total of 52 (32.1%) of the 162 contacted patients returned the questionnaire; 10 (19.2%) patients failed to add a photograph. The median follow-up of the survey participants was 59.5 months (IQR 42–132 months).

The survey results, including sections regarding wound healing, complications, scar pain, and imaging as well as satisfaction regarding the scar and the operative procedure, are summarized in Table 5. All patients confirmed the usage of resorbable sutures. A total of 5 parents reported the presence of scar hernia, which could only be verified in 2 patients in the clinical examination.

Satisfaction regarding cosmesis was high. Seventy-five percent of the parents and 87.2% of the patients were very satisfied/satisfied with the appearance of the scar. Only 4 (7.7%) parents and 2 patient (4.2%) were dissatisfied/very dissatisfied. A majority of 39 (78%) parents would decide again for an open pyeloplasty.

The median scar length was 4.0 cm (range: 1.9–7.1 cm). Both surgeons categorized the scars in 82.2% with a scar scale score of 0–2. The overall consistency regarding vascularization, pigmentation, and level of the scar was between 76.2% and 90.5%. Our observed low scar scale score was in accordance with the parents’ assessments. Detailed data are shown in Table 6.

## 4. Discussion

A successful pyeloplasty, regardless of the type of surgery, should result in an improved renal function and hydronephrosis on ultrasound as well as in a long-term cosmetical satisfaction. Currently, there are three main approaches of performing pyeloplasty (open, laparoscopic, robot-assisted). In the last years, several studies showed better outcomes regarding laparoscopic and robotic-assisted procedures in pediatric urology [3,9,12,22,23]. In consequence, there seems to be a paucity concerning the feasibility of open pyeloplasty. However, there is wide acknowledgement that minimal invasive surgery in infants under 1 year and under 10 kg of weight is still a challenging approach [22]. Therefore, the superiority of open pyeloplasty in this special population is still a matter of debate.

### 4.1. Perioperative Management

In 2006, Lee et al. already demonstrated that RALP can be performed in infants, young children, and teenagers with similar success. They compared RALP with an age-matched cohort of patients undergoing open pyeloplasty. The operative times were significantly higher in the RALP procedure (219 min in RALP vs. 181 min in open pyeloplasty); however, with increasing experience, RALP operative times improved and approached the open times of three experienced surgeons [16].

Currently, Rague et al. reported an operating time (cut-close time) of 200 min for RALP and 165 min for open pyeloplasty [9]. Only Kafka et al. showed a lower mean operative time for RALP with 67.8 min and 66.5 min for open pyeloplasty [3].

In a recently published systematic review and meta-analysis, Cascini et al. investigated nine studies. They compared MIS, including laparoscopic pyeloplasty (LP) and RALP, with open pyeloplasty. No significant differences were found concerning age, weight, complications, and failure rate. Operative time was significantly lower in open pyeloplasty than that in MIS. They also stated that LP showed a lengthier learning curve than RALP; moreover, they reported that the learning curve for RALP seemed to be comparable to the one for open pyeloplasty. Nevertheless, the robotic approach is challenging in infants because of limited space for port placement, the absence of correct-sized trocars, and restricted working space. However, a laparoscopic approach using 3 mm instruments is safe and feasible and has been adopted by many centers for years now [24].

In our study, we reported a median operation time of 106 min independent of the level of training. Therefore, we demonstrated again the lower operation time for open pyeloplasty in comparison with the generally measured operation time in RALP [12].

Additionally, our study shows that regarding open pyeloplasty, we are rather in the lower range of operative time despite being a surgical training center for pediatric urology. Especially, the comparable operative times between fellow and experienced surgeon demonstrate that surgical education can be performed without severe burden for infants. Andolfi et al. suggested that an experience of at least 25–50 RALP procedures in older patients is necessary before approaching an infant [25].

### 4.2. Complications

Complication rates vary between 3.4% and 31.6%, depending on the published series, with a tendentially lower complication rate in open pyeloplasty [12,23,26].

Surprisingly, the overall complication rate in our population was 27.7%, which is comparatively high. One reason might be the different rating of the period of complication occurrence. In our study, we summarized all intra- and postoperative complications and complications in the long-term follow-up.

Regarding intraoperative complications, failure of ureteral stent insertion was most common. Similar results have been shown in the literature regardless the type of pyeloplasty [23].

While performing laparoscopic pyeloplasty or RALP, conversion might be an additional intraoperative complication. The current literature shows that the need for conversion is generally rare [23]. This might carry a high risk that an open pyeloplasty can no longer be performed safely especially in an emergency case.

In the case of postoperative complications, the most severe complications in our study were CD 3b. Most of them are stent dislocations requiring anesthesia for replacement. This may lead to an upscaled Clavien–Dindo grading in pediatric pyeloplasty [12]. The most significant major complication according to CD 3b was 1 (0.6%) redo pyeloplasty, which was lower than the reported reoperation rate (4.8%–5.1%) in a meta-analysis reported by Cundy et al. [12]. They showed a similar overall success rate regarding robot-assisted versus conventional laparoscopic and open pyeloplasty in children. Reoperation rates were lower in RALP versus laparoscopic and open pyeloplasty but without statistical significance [12]. He et al. reported a success rate of 94.2% with a mean follow-up of 12 months in their study about laparoscopic pyeloplasty [23]. Other studies showed similar success rates of 97% in RALP groups vs. 96% in groups with open pyeloplasty [13]. At least in our study, we achieved a success rate of 99.4%.

In general, febrile urinary tract infections are the overall most common complication (CD 2). He et al. reported in 2020 a rate of 10% in his population, which is comparable to our findings [23]. They usually occur with the indwelling ureteral stent; therefore, they are the most common reason for the readmission within the first 30 days after surgery.

It is noticeable that the few existing studies about RALP in infants under 1 year mainly focus on surgical complications and outcome. There is a gap regarding anesthesiologic challenges and complications particularly due to pneumoperitoneum [25]. These challenges may result in a restriction of RALP to high-volume centers particularly in children under 1 year.

### 4.3. Length of Hospital Stay

In several studies, the length of hospital stay is one of the investigated factors to prove the effectiveness of the respective procedure. With a mean of 5.2 days, the length of stay in our study was much longer than the reported ones for RALP (1.4–2.3 days) [11,16,25], as well as for open pyeloplasty (2–3.5 days) [11,16]. This might be caused by the possibility of our healthcare system of keeping infants inpatient as long as they are provided with a transurethral catheter. However, regarding the postoperative need for analgesia, which was in median required for 42 h (minimum of 2 h, maximum of 480 h), a discharge might be possible after 2 days. Therefore, our findings would be similar to the previously reported length of hospital stay in the literature.

### 4.4. Cosmesis and Satisfaction

One major argument for the recent increase in RALP is the improved cosmesis results. In 2013, Barbosa et al. investigated parents and patients’ perception of robotic vs. open urological scars in children. They showed that parents and patients preferred robotic scars. Nevertheless, it seems that the majority of parents would ultimately base their choice of surgical approach on clinical efficacy rather than scar preference [14]. We can confirm this statement. In our study, 77.3% of the parents would choose open surgery again, even when they were dissatisfied with the scar.

To objectify the result from the questionnaire, the current appearance of the scar was assessed on a photograph. In general, the Patient Scar Assessment Questionnaire (PSAQ) as a patient-based assessment tool for linear scars is used [27]. We decided to use a solely examiner-based modified Vancouver Scar Scale to achieve a higher return rate [20,28,29]. Most scars were graded with a low score, which is a favorable result. This is also reflected in the high level of parent satisfaction.

### 4.5. Limitations

Our monocentric study was limited by the lack of a control group and the partially retrospective design. Additionally, due to a consistent lack of data, an objective evaluation of postoperative pain was not possible. Furthermore, the response rate of the questionnaire was low, and we did not use a standardized questionnaire, which made our data solitary and noncomparable with other studies. Despite these limitations, we were able to evaluate a significant number of patients.

## 5. Conclusions

Open pyeloplasty is often presented as an inferior approach in comparison with laparoscopic pyeloplasty and RALP, especially in terms of reduced postoperative pain, low complication rates, better cosmetic results, and shortened hospitalization. Despite all the positive reports for MIS, there are still many hurdles to overcome. Especially in infants under 1 year of age, MIS is mostly limited to high-volume centers with experienced surgeons. Until these problems are solved or improved, open pyeloplasty should be considered a safe and feasible procedure with good cosmetic outcome that any pediatric urologist or surgeon can safely perform.

## Figures and Tables

**Figure 1 children-10-00257-f001:**
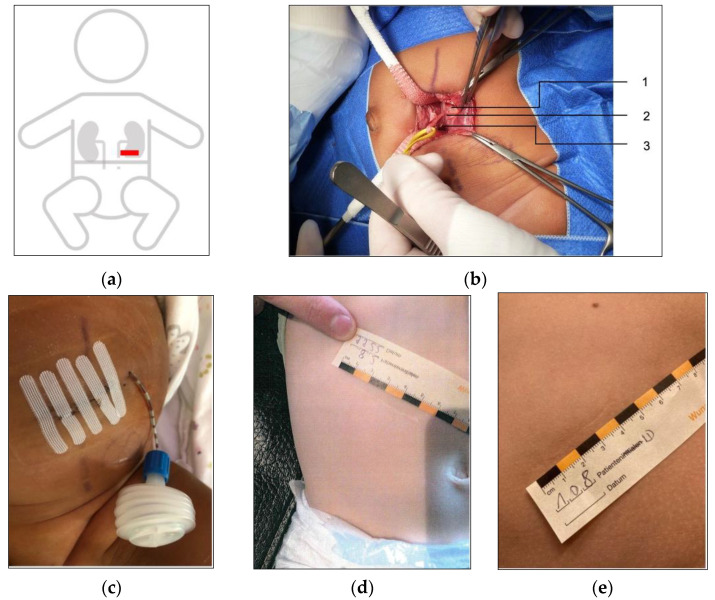
(**a**) Incision; (**b**) intraoperative view: (1) renal pelvis, (2) ureteropelvic junction obstruction, and (3) ureter; (**c**) postoperative wound; (**d**) scar after 22 months; and (**e**) scar after 12 years.

**Table 1 children-10-00257-t001:** Demographic and clinical characteristics.

Age (median (min–max))	3 months (0–11 months)
Gender male/female (%)	124 (76.5%)/38 (23.5%)
Weight (mean ± SD)	6570 g (SD ± 1496 g)
Affected side (left/right)	89 (54.9%)/72 (44.4%)
Differential renal function < 45% (*n* = 151)	70 (46.4%)
Hospitalization (mean ± SD)	5.2 days (SD ± 1 days)
Follow up (median (min–max))	30.5 months (0–162 months)

**Table 2 children-10-00257-t002:** Surgical data and complications.

Operation time (median (min–max)	106 min (63–198 min)
With retrograde urography (no. of patients)	12 (7.4%)
Without retrograde urography (no. of patients)	150 (92.6%)
Urinary drainage—ureteral stent (no. of patients)	155 (95.7%)
Urinary drainage—nephrostomy (no. of patients)	6 (3.7%)
Duration of urinary drainage (mean ± SD)	41.4 days (SD ± 9.8 days)
Readmission within 30 days	19 (11.7%)
Overall complications	45 (27.7%)
Intraoperative Complications	7 (4.3%)
Postoperative complications	36 (22.2%)
Long-term complications	2 (1.2%)

**Table 3 children-10-00257-t003:** Postoperative complications.

Clavien–Dindo Score	Diagnosis/Treatment	Number of Patients	Total
1	Intraureteral stent dislocation, use of URS for removal	1	3 (1.8%)
	Stent dislocation, no new stent	1	
	Urinom, stent stays longer than usual	1	
2	Febrile urinary tract infection		16 (9.8%)
3a	(-)		(-)
3b	Stent dislocation with restenting	11	17 (10.5%)
	Change nephrostomy to JJ stent	2	
	Clogged stent	1	
	Stent replacement in UTI	1	
	Urinom, placement of JJ stent	1	
	Redo pyeloplasty	1	

**Table 4 children-10-00257-t004:** Preoperative ultrasound findings and follow-up.

	Preop	Postop 1 Month	Postop 6 Months	Postop 12 Months	Postop 5 Years
Renal pelvis diameter (APD) (no pat.)	22.0 mm (IQR 17–30) (*n* = 157)	19.0 mm (IQR 13–24) (*n* = 140)	9.5 mm (IQR 7–14) (*n* = 108)	8.0 mm (IQR 4–12) (*n* = 102)	6 mm (IQR 2–12) (*n* = 47)
Parenchymal thickness (no pat.)	4.9 mm (IQR 3.8–6.0) (*n* = 130)	(-)	(-)	9.0 mm (IQR 7–11) (*n* = 46)	11.0 mm (IQR 10–14) (*n* = 26)
Calyceal diameter (no pat.)	13.0 mm (IQR 10–18) (*n* = 121)	11 mm (IQR 8–15) (*n* = 77)	6 mm (IQR 0–9) (*n* = 42)	4 mm (IQR 0–9) (*n* = 47)	0.0 mm (IQR 0–5.2) (*n* = 30)

**Table 5 children-10-00257-t005:** Questionnaire results.

Question	Responses	Results
How did the wound heal after surgery?	No complications	44 (84.6%)
	Slight redness, no pain, wound remained closed	8 (15.4%)
	Significant redness and infection, pain, wound remained closed	0
	Wound was inflamed and secreted, wound has opened superficially, no reoperation required	0
	Wound was inflamed and secreted, wound has opened, further surgery for wound closure was necessary	0
Were there any further complications after the surgery?	None	39 (75%)
	Strong pain	2 (3.8%)
	Bleeding	0
	Urinary tract infection	5 (9.6%)
	Wound infection	1 (1.9%)
	Hernia formation (formation of a gap/protrusion in the area of the scar)	5 (9.6%)
	Other	0
Is your child currently still complaining of pain in the area of the scar?	Yes	2 (3.8%)
	No	50 (96.2%)
How does the scar currently look?	Fine white line, barely visible	15 (29.4%)
	White line, visible, flat, and level	28 (54.9%)
	White line, visible, raised	67 (13.7%)
	Red, raised	0
	Bulging	1 (2%)
	Missing	1 (1.9%)
How satisfied are you (parents) with the appearance of the scar?	Very satisfied	24 (46.2%)
	Satisfied	15 (28.8%)
	Okay	9 (17.3%)
	Dissatisfied	3 (5.8%)
	Very dissatisfied	1 (1.9%)
How satisfied are you (patient) with the appearance of the scar? (Please let your child answer.)	Very satisfied (the scar does not bother me at all)	33 (70.2%)
	Satisfied (the scar hardly bothers me)	8 (17%)
	Okay (well, the scar bothers me occasionally)	4 (8.5%)
	dissatisfied (the scar bothers me)	1 (2.1%)
	Dissatisfied (the scar bothers me and I try to hide it)	1 (2.1%)
	Missing	5 (9.6%)
Were you satisfied with the surgery overall (preparation, surgery, and inpatient care)?	Very satisfied	30 (57.7%)
	Satisfied	20 (38.5%)
	Okay	1 (1.9%)
	Dissatisfied	1 (1.9%)
	Very dissatisfied	0
Would you have the operation performed again as an open pyeloplasty?	Yes	39 (75%)
	No	11 (21.2%)
	Missing	2 (3.8%)

**Table 6 children-10-00257-t006:** Scar assessment by surgeons.

	Surgeon 1	Surgeon 2	Consistency
Vascularization:			90.5%
Normal	38 (90.5%)	40 (95.2%)	
Rose	4 (9.5%)	2 (4.8%)	
Red	0	0	
Purple	0	0	
Pigmentation:			88.1%
Normal	20 (47.6%)	19 (45.2%)	
Hypopigmented	22 (52.4%)	21 (50%)	
Hyperpigmented	0	2 (4.8%)	
Level:			76.2%
Flat/level	32 (76.2%)	32 (76.2%)	
Slightly raised (<2 mm)	3 (7.1%)	3 (7.1%)	
Fairly raised (2–5 mm)	0	2 (4.8%)	
Bulging (>5 mm)	0	1 (2.4%)	
Sunken	7 (16.7%)	4 (9.5%)	

## Data Availability

The data used to support the findings of this study are available from the corresponding author upon request.

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
