# Peer review of "Open Pyeloplasty in Infants under 1 Year—Proven or Meaningless?"

_children, 2023, doi:10.3390/children10020257_

Round 1

Reviewer 1 Report

Title: Open pyeloplasty in infants under 1 year – does it still have a raison d’être? - being an english language article, we should keep the english words, except the global accepted terminology as latin word/nomina anatomica etc

Different other phrases that could be rephrased: e.g: „Nowadays open pyeloplasty (OP) seems to be out of fashion. ”, „Excluded were patients with other 57 cause of hydronephrosis than uretero-pelvic junction obstruction”

Reviewer 2 Report

Thanks to Children for inviting me to review this manuscript.

The Authors show a retrospective evaluation of 162 infants treated with open pyeloplasy.

My first subjective and absolutely personal consideration is that the surgical approach and technique proposed are still actual and valid. Therefore, since the actual trend (as also the Authors exposed) is the progressive diffusion and widespread of minimally invasive surgery (especially robotic), its limits let us reconsider the "older" open surgery approach.

The Authors present an high number of patients, they evaluate many aspects through the objective and subjective points of view (from the patients' families one and from the surgeons one).

The topic is in countertrend. The methodology is robust. The layout style is good. Language and english style are correct. Images are appropriate. References are appropriate.

To improve some minor critical issues, I would like to offer some suggestions and I hope the Authors would agree with me. If not, I expect for an appropriate explanation.

Line 10: maybe, if the expression "out of fashion" was written in quotes, it could fit better in the sentence;

Lines 39-42: it could be necessary to reformulate this sentences, since there are some linguistic imperfections;

Line 46: I suggest to add a bibliographic source.

Line 47: I disagree with the sentence "as operative time decrease, operative costs decrease as well". In particular, the operative time in a robotic procedure implicates higher costs than in the laparoscopic or open techniques not only and directly dependent on the duration. The robotic surgical instruments are more expensive than the open or laparoscopic correspondents. The time improving in robotic surgery does not directly correlate with a cost reduction.

Please, reformulate the sentence or motivate it with some sources.  

Line 74: I suggest to specify when the second part of the study was conducted. In addiction, explain how the survey was conducted (mail, telephone, outpatient clinic, etc). 

Line 111: Did you color the urine with methylene blue or similar agents to improve the reflux visualization?

Line 128: I suggest to describe a little deeper which antibiotic prophylaxis was received by the patients. If the data is not available, did you refer to EAU or other guidelines, or did you follow an internal protocol?

Line 131-136: was post-operative ultrasonography follow up performed by the radiologist or urologist/pediatric surgeon? The addition of this detail would confer more strength to internal validity and reduce theoretical bias.

Line 334-338: First, I suggest to dedicate a specific paragraph to limits explanation.  

I personally think that further limits of this study are its monocentric nature and the lack of pain objective evaluation (for example Flacc Scale Merkel S, Voepel-Lewis T, Shayevitz JR, et al:The FLACC: A behavioural scale for scoring postoperative pain in young children. Pediatric nursing 1997; 23:293-797) in the patients involved. I suggest to declare them in the text.

Best Regards

Author Response

please see the attachement

Reviewer 3 Report

The authors present an impressive (162 cases) retrospective analysis of open pyeloplasties in children under the age of 1 year. They asses the therapeutic outcomes, the cosmetics and patients satisfaction after this procedure. The introduction provides enough information. The study designed is well planned and the methodology well conducted.

 Line 98 – Where was the catheter inserted? Is it a bladder catheter? It should be clearly specified.

Line 178 – 180 - `What do the author mean by urinary drainage? Did they mean external drainage, JJ stent or bladder catheter?

Line 252. Perioperative management

There is no mention in this paragraph concerning the laparoscopic approach in infants. The robotic approach is challenging in small children mainly because of the size of the robotic instruments. However, laparoscopic approach using 3mm instruments is safe and feasible and has been adopted by many centres for years now (See for instance: Cascini V, Lauriti G, Di Renzo D, Miscia ME, Lisi G. Ureteropelvic junction obstruction in infants: Open or minimally invasive surgery? A systematic review and meta-analysis. Front Pediatr. 2022 Nov 23;10:1052440.)

Line 272. Complications rate is rather high. Please provide more explanations/ discussions for this aspect.

Line 309 – Length of hospital stay is rather high compared with the international standards. What was the reason for that? Why was the reason to keep the bladder catheter in place for such long time? After RALP or LP, the catheter may be removed the next day.

Line 314 – A more useful approach would be to compare the need for analgesia.

Line 320 - Cosmesis and satisfaction

Line 325 – Off course, most of the parents were satisfied with the result? Their children are well now.  The clinical outcome is the main goal of the treatment

Overall, the manuscript give the impression of a plea in favour of the OP versus the minimal invasive approaches, mainly in the discussion section. The reader might be misled to believe that OP is superior to the other approaches in all aspects. This is false. OP has its place in the operative treatment of UPJO and any surgeon whom deal with UPJO should be prepared for an OP. Minimal invasive pyeloplasty is indeed challenging in infants, but is not impossible and should be preferred whenever possible. The advantages of minimal invasive approach are clear and well known nowadays. Considering this I recommend an extensive review of the discussion and conclusions section to a more “neutral” approach.

Round 2

Reviewer 3 Report

Dear authors

Thank you for replying to my comments and for updating the manuscript.

I would incorporate this statement (your statement) or at least part of it in the discussion and/ or conclusion sections: 

„This manuscript should not be a plea in favour for open pyeloplasty, but rather a reminder that despite all the enthusiasm and all the positive reports so far for MIS, especially RALP, there are still many hurdles to be overcome. Until the problems are solved or improved, open pyeloplasty should not be forgotten as a safe alternative and should be a procedure that any pediatric urologist or pediatric surgeon can safely perform.“

This would clear up your intention with this manuscript and remove the feeling of  tendentiousness.  

I have no further comments. 

Author Response

Dear Reviewer 3,

thank you for your suggestions. We think they will make the intentions of the manuscript clearer. We added the statement into the conclusion. 
